# Analyzing the Renewable Energy and CO$_2$ Emission Levels Nexus at an EU Level: A Panel Data Regression Approach

**Mihail Busu *** and **Alexandra Catalina Nedelcu**

Faculty of Business Administration in Foreign Languages, The Bucharest University of Economic Studies, 6 Piata Romana, 1st District, 010374 Bucharest, Romania; catalina.nedelcu@fabiz.ase.ro
\* Correspondence: mihail.busu@fabiz.ase.ro

**Abstract:** In the past decades, carbon dioxide (CO$_2$) emissions have become an important issue for many researchers and policy makers. The focus of scientists and experts in the area is mainly on lowering the CO$_2$ emission levels. In this article, panel data is analyzed with an econometric model, to estimate the impact of renewable energy, biofuels, bioenergy efficiency, population, and urbanization level on CO$_2$ emissions in European Union (EU) countries. Our results underline the fact that urbanization level has a negative impact on increasing CO$_2$ emissions, while biofuels, bioenergy production, and renewable energy consumption have positive and direct impacts on reducing CO$_2$ emissions. Moreover, population growth and urbanization level are negatively correlated with CO$_2$ emission levels. The authors' findings suggest that the public policies at the national level must encourage the consumption of renewable energy and biofuels in the EU, while population and urbanization level should come along with more restrictions on CO$_2$ emissions.

**Keywords:** renewable energy; CO$_2$ emissions; biofuels; bioenergy; panel data





## 1. Introduction

The use of sustainable energy produced by renewable energy sources (RES) in European countries in order to achieve the objectives set by the European Union (EU) implies the need for data collection, which relates to knowledge of the RES, government policy, and economic interests. This is the starting point for a detailed analysis to determine the actual stage in the transition to clean energy process [1,2]. Economic modelling is the next step in the contextual conversion plan to respond to the predictions recommended by decision makers [3]. Research activities in the clean energy field will significantly contribute to the observation of renewable energy production limits, as well as to the identification of possible barriers to the development of this industry [4]. In the past two decades, intensive research was conducted to analyze the challenges and barriers for energy production using RES, most of them being linked to its costs, transport, and storage, or political and regulatory issues [5–7]. Although many European countries have made significant progress in developing RES capacities, there is still important potential to be explored to meet the targets imposed by EU.

Regarding the development of clean energy at the national level, we observed the potential generated by certain industrial branches, specific to the basic sectors, to produce bioenergy and biofuels. This also includes the bio-industry (i.e., biofuel production and biodegradable plastic products), as well as the food industry and the other tertiary sectors that are interdependent on these two sectors, such as the pharmaceutical and chemical industries [8].

The available energy in the EU relies on the energy imported from other non-EU member states and the energy produced by the EU member states. In 2018, the EU countries produced around 45% of the energy they consumed, while they imported about 55%. In Figure 1, it can be observed the percentages of the energy produced in the EU, by source of energy.

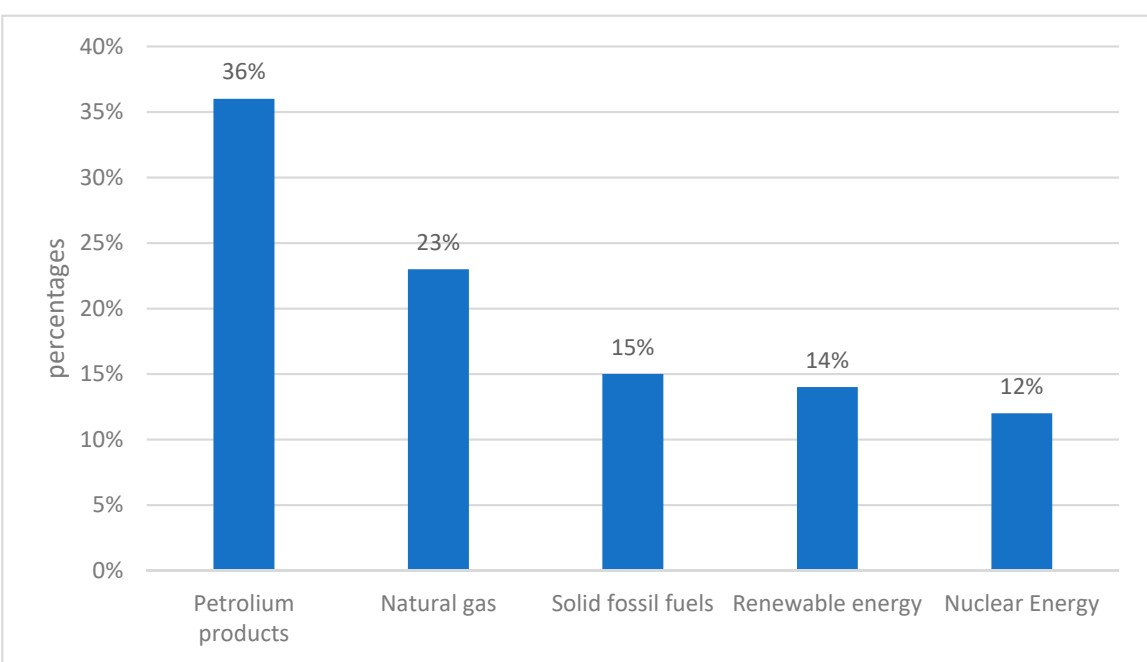

**Figure 1.** Energy produced in the EU in 2019, by source of energy. Source: authors' own computations on the data collected from European Union Statistical Office (Eurostat) [9].

From the above figure we could observe that the energy produced in the EU member states came from petroleum products (36%), followed by natural gas (23%), solid fossil fuels (15%), RES (14%), and nuclear energy (12%). According to Eurostat [9], the RES level is still below the 2020 target (20%) and 2030 target (32%).

Figure 2 reveals the consumption of the RES, at the EU level, as a percentage from total energy consumed, in 2018.

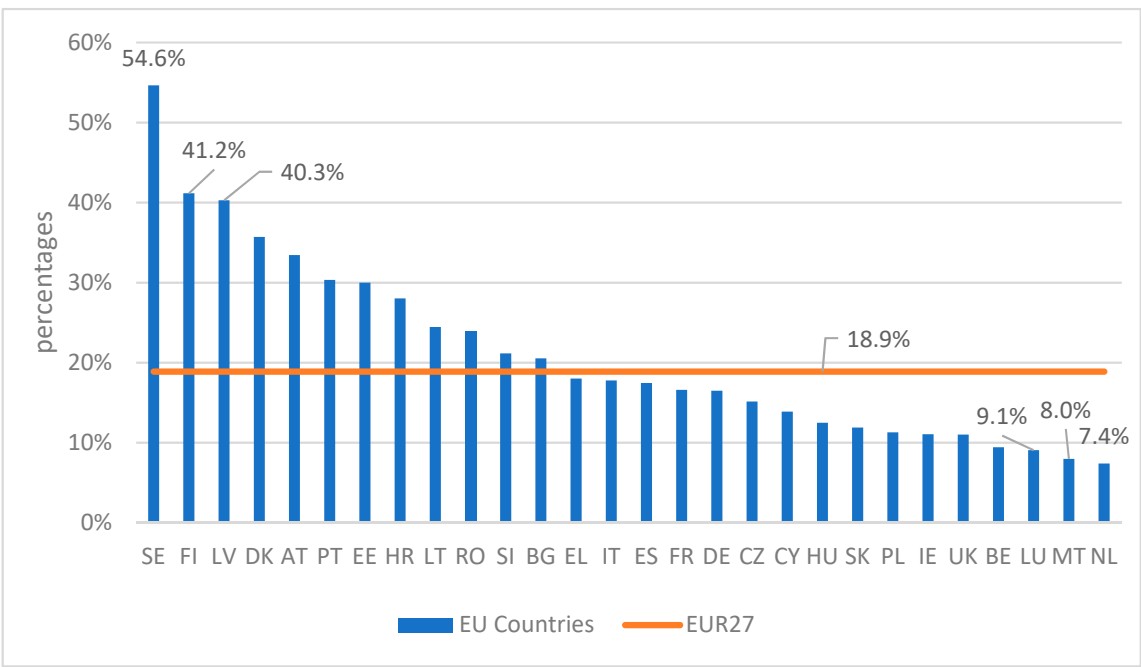

**Figure 2.** The shares of renewable energy sources (RES) in total energy consumption in EU member states, in 2019. Source: authors' own computations on the data collected from Eurostat [9].

In the above figure, we can see that the EU member states with the highest level of RES consumption in 2018 were Sweden (54.6%), Finland (41.2%), and Latvia (40.3%), while the EU member states with the lowest levels of renewable energy consumption were Luxembourg (9.1%), Malta (8.0%), and the Netherlands (7.4%).

The electricity generation sector is mainly based on the type of primary resource used in the electricity production process (i.e., thermal, nuclear, hydro, coal, wind, biomass, and photovoltaic).

In Figure 3 we present the structure of the electricity delivered from the dispatchable production units, calculated based on renewable and conventional resources.

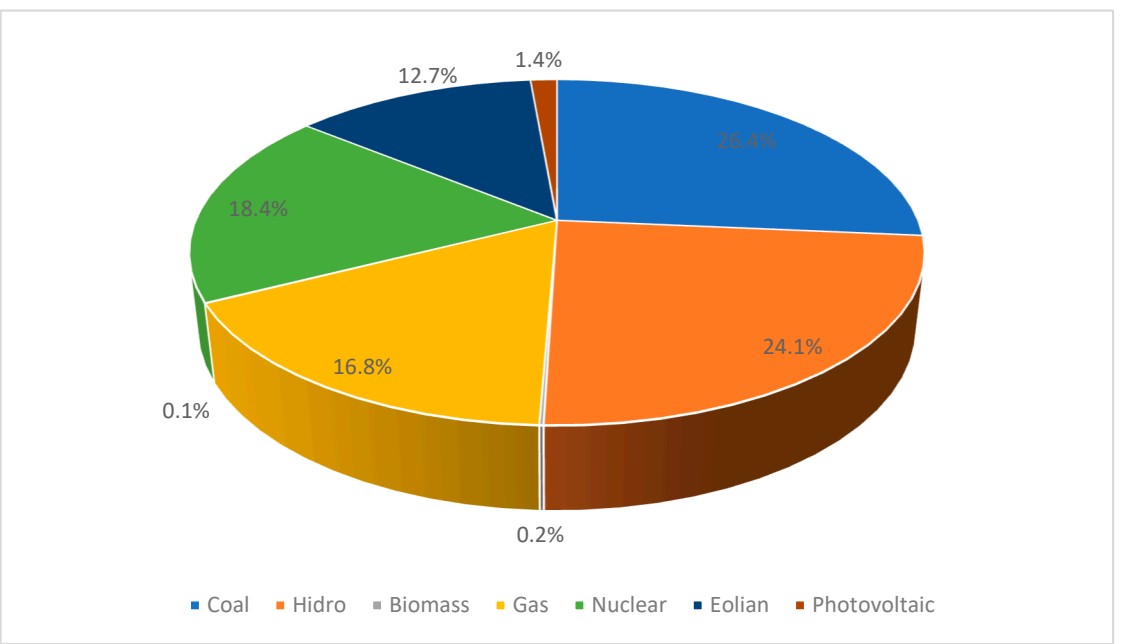

**Figure 3.** The structure of the energy resource shares in EU, in 2019. Source: authors' own computations on the data collected from Eurostat [9].

The aim of this research is to model the impact of biofuels, RES, bioenergy, population, and urbanization levels on the emissions of carbon dioxide in EU member states, between 2000 and 2019, to substantiate the national plan needed to achieve the EU targets. To perform this analysis, an econometric study was conducted using an econometric model analyzed with the help of statistical software Econometric Views (EViews) 11.

This manuscript includes a brief literature review followed by the hypotheses' development. Going forward, the evolution of the model indicators is presented, then an econometric analysis is performed and the parameters of the regression equation are estimated. The research assumptions are presented and tested next. In the Conclusion chapter are summarized authors' recommendations, limitations of this study, and future research implications.

## 2. Literature Review and Hypotheses Development

The targets set by the EU, both for RES production and for reduction of $CO_2$ emissions, are well known, but are they linked? Many researchers addressed this question in previous studies on the $CO_2$ emissions and RES nexus.

It was demonstrated that there is a strong connection between $CO_2$ emissions, RES, and urbanization level [10,11]. Other authors [12,13] argue that, at the EU level, bioenergy and RES have a positive and strong impact on $CO_2$ emissions.

Moreover, while some scientists [14–16] concluded that the use of bioenergy and biofuels is strongly correlated with the reduction of carbon dioxide emissions, other authors [17–19] argue that population level and RES are strongly correlated with $CO_2$ emission

levels. It was also proved [20–22] that biofuels and urbanization level have a positive and strong influence on lowering $CO_2$ emissions levels. Nevertheless, other economists [23,24] demonstrated that population growth and urbanization rate have a strong and indirect influence on lowering carbon dioxide emission levels.

The $CO_2$ emissions and RES nexus in EU was analyzed by other researchers in the past decades. Ciupageanu et al. [25] argue that economic and population growths as well as urbanization are significant factors of increasing levels of $CO_2$. While some authors [26] consider that investing in biofuels and bioenergy are key factors for a low carbon society, other studies [27,28] demonstrate that $CO_2$ emissions are highly correlated with urbanization, economic growth and renewable energy consumptions. Other researchers [29] argued that, at a European level, there is still room for improvement in terms of management practices in the energy sector, and the $CO_2$ emission levels are very high.

The impact of RES on $CO_2$ emissions was analyzed by many researchers. Sims et al. [30] analyzed the differences in the cost of electricity between present and future technologies which will be available in the next decade. The authors consider that the new technologies would reduce $CO_2$ emissions by 14% until 2020. Other authors [31] analyzed the causality relationship between nuclear energy, RES, and $CO_2$ emissions in the USA from 1960 to 2007, without reaching a conclusion regarding the impact of the analyzed variables on $CO_2$ emission reduction.

At the EU level, Kök et al. [32] concluded that an increase of investment in RES would lead to a consistent reduction of $CO_2$ emissions. Other researchers [33] argued that a method of financing projects based on reducing levels of $CO_2$ emissions is by selling carbon revenue bonds at ten years maturity.

All these articles reveal that while some factors, such as population, economic growth, and urbanization rate could increase $CO_2$ levels, energy productivity and RES are useful factors for decarbonization at the European Union level. All the above results will be the basis of our statistical hypotheses developed in the next section.

Nowadays, one of the most important goals of our society is to reduce $CO_2$ emission levels, thus to develop RES capacities, and to increase consumption of clean energy seems to be one of the best solutions.

Due to data availability, in our analysis, we will start our research from the year 2000. The literature review references helped us to formulate our research question: "Which are the clean energy factors that impact on $CO_2$ emissions in the EU?" Then, we will further continue our study by making a multiple regression estimation to analyze the impact of the exogenous variables on the endogenous variable.

Thus, we will develop and test the six statistical hypotheses presented in Table 1.

**Table 1.** Statistical Hypotheses of the Regression Model.

| Hypotheses | |
| --- | --- |
| $H_1$ | Renewable energy sources (RES) are highly correlated with $CO_2$ emissions. |
| $H_2$ | Biofuels are a significant factor in $CO_2$ emissions. |
| $H_3$ | Bioenergy productivity is strongly correlated with $CO_2$ emissions. |
| $H_4$ | Population is a significant factor in $CO_2$ emissions. |
| $H_5$ | Urbanization is a significant factor in $CO_2$ emissions. |
| $H_6$ | Real Gross Domestic Product (GDP) per capita is a significant factor in $CO_2$ emissions. |

All six hypotheses shall be analyzed with an econometric model, which will be presented in the following section.

## 3. Materials and Methods

The panel data multiple regression model is an econometric model with six exogenous factors: RES, biofuels, bioenergy, urbanization level, and population. These are representative factors for the dependent variable in our analysis, the carbon dioxide emission levels in the EU.

A description of the variables of the econometric model can be observed in Table 2. The data were collected from Eurostat for all EU member states, 2000 and 2019. The panel data regression model was estimated with the statistical software EViews 11.

**Table 2.** A description of the variables of the econometric model.

| Variable | Name | Definition | Unit |
|---|---|---|---|
| (Y) | $CO_2$ emissions | The total $CO_2$ emissions | Millions of tones |
| ($X_1$) | Renewable energy | Renewable energy consumed divided by total energy | Percentages (%) |
| ($X_2$) | Biofuels | Biofuels production | Thousand tones |
| ($X_3$) | Bioenergy productivity | GDP divided by the gross inland consumption of bioenergy in one year | Euro/kg |
| ($X_4$) | Population | Population of each EU country | Millions |
| ($X_5$) | Urbanization | Urban population share from total population | Percentages (%) |
| ($X_6$) | Real GDP per capita | The rate of the Real GDP per capita in EU countries, and the number of inhabitants | Millions of euro |

The econometric model uses a panel dataset. The selection of the sample and the results from the data analysis are consistent [34,35]. Thus, we could conclude that the sample size used in our analysis is significant.

Therefore, the econometric model is described in Equation (1):

$$y_i = \beta_0 + \beta_1 (x_1)_{it} + \beta_2 (x_2)_{it} + \beta_3 (x_3)_{it} + \beta_4 (x_4)_{it} + \beta_5 (x_5)_{it} + \beta_6 (x_6)_{it} + \varepsilon \quad (1)$$

where,

- (y)—dependent variable
- $x_1, x_2, x_3, x_4, x_5,$ and $x_6$—independent variables
- $\beta_0, \beta_1, \beta_2, \beta_3, \beta_4, \beta_5,$ and $\beta_6$—parametric coefficients
- i—1,..,28—the number of countries; t—1,..20—time frame
- $\varepsilon$—error term

In the econometric model will be used the previously described endogenous and exogenous variables which will lead to the following econometric equation:

$$(CO_2 \text{ emmisions})_i$$
$$= \beta_0 + \beta_1 (\text{Renewable})_i + \beta_2 (\text{Biofuels})_i + \beta_3 (\text{Bioenergy})_i + \beta_4 (\text{Population})_i \quad (2)$$
$$+ \beta_5 (\text{Urbanization})_i + \beta_6 (\text{Real GDP per capita})_i$$

where,

- $CO_2$ emissions—the total levels of $CO_2$ emission in EU countries
- Renewable—the rate of RES in total energy
- Biofuels—production of biofuels
- Bioenergy—bioenergy productivity
- Population—represents the total population in EU countries
- Urbanization—the urbanization degree in total population
- Real GDP per capita—real GDP divided by the number of inhabitants

The data used in our analysis for the variables above were collected from Eurostat.

Further, Hausman test will be used to test whether a random effect or a fixed effects model will be used. The correlation between the exogenous variables and the error term is tested. The Null and Alternative Hypotheses are:

Ho: The model has random effect: $E(X_{it}/\varepsilon_{it}) = 0$

Ha: The model has fixed effects: $E(X_{it}/\varepsilon_{it}) \neq 0$

The Hausman statistic is calculated with the formula:

$$H = (\hat{\beta}_{FE} - \hat{\beta}_{FE})^T (var(\hat{\beta}_{FE}) - var(\hat{\beta}_{RE}))^{-1} (\hat{\beta}_{FE} - \hat{\beta}_{RE}) \quad (3)$$

where $\hat{\beta}_{RE}$ is the random effects model estimator and $\hat{\beta}_{FE}$. is the random effects model estimator.

Hausman test will be performed to choose between a fixed effects (FE) model and a random effects (RE) model.

To test the collinearity between the exogenous variables, the variance inflection factor (VIF) test was used. The VIF test is given by the following formula:

$$\mathbf{VIF} = \frac{1}{1 - \mathbf{R}_j^2} \tag{4}$$

where $\mathbf{R}_j^2$ is the regression coefficient of determination.

The coefficient of determination is defined as the percentage of the variability of the dependent variable explained by the econometric model.

The VIF value reflects the uncertainty in the coefficient estimates. If the VIF value is close to 10, we conclude that the exogenous variables are correlated, while a value close to 1 underlines the fact that the independent variables are not corelated. A threshold value of 5 is usually used.

In the next section, the results will be discussed and analyzed.

## 4. Results and Discussions

We have used the multiple regression method to estimate the econometric model. The six statistical hypotheses described above were also tested through this methodology. The panel data linear regression equation evaluates a multivariable function of the dependent variable to analyze the independent variables.

In Table 3 are presented the factors used in the panel data regression model.

**Table 3.** Descriptive Statistics.

| Variable | Mean | Median | Standard Deviation | N |
|---|---|---|---|---|
| $CO_2$ emissions (Y) | 68.6 | 68.2 | 13.18 | 560 |
| Renewable ($X_1$) | 21.7 | 22.5 | 3.02 | 560 |
| Biofuels ($X_2$) | 3.51 | 3.64 | 0.35 | 560 |
| Bioenergy ($X_3$) | 3.32 | 3.4 | 0.78 | 560 |
| Population ($X_4$) | 22.52 | 22.52 | 0.25 | 560 |
| Urbanization ($X_5$) | 53.6 | 53.8 | 0.45 | 560 |
| Real GDP per capita ($X_6$) | 5.68 | 6.35 | 2.03 | 560 |

Source: Authors' own calculations using EViews 11.

The correlation matrix between the indicators of the econometric model is presented in Table 4. The correlation matrix enables us to verify whether the model has multicollinearity issues. According to Dabholkar [36], the independent factors are not correlated when the correlation coefficients have values within ±0.30.

**Table 4.** The Correlation Matrix.

| Variable | Y | $X_1$ | $X_2$ | $X_3$ | $X_4$ | $X_5$ | $X_6$ |
|---|---|---|---|---|---|---|---|
| Y | 1 | - | - | - | - | - | - |
| $X_1$ | 0.714 | 1 | - | - | - | - | - |
| $X_2$ | 0.689 | 0.098 | 1 | - | - | - | - |
| $X_3$ | 0.702 | 0.134 | 0.078 | 1 | - | - | - |
| $X_4$ | 0.544 | 0.198 | 0.105 | 0.102 | 1 | - | - |
| $X_5$ | 0.612 | 0.233 | 0.099 | 0.105 | 0.112 | 1 | - |
| $X_6$ | 0.623 | 0.203 | 0.104 | 0.125 | 0.107 | 0.188 | 1 |

Source: Authors' determined values by using EViews 11 software package.

In the table above, we could observe that there is high correlation between the dependent variable and independent variables, while the exogenous variables are low correlated. Hence, we could conclude that the independent variables are not correlated. Additionally,

we could see that there is high correlation between $CO_2$ emission levels and RES (71.4%) and energy productivity (70.2%), while renewable energy and urbanization (23.3%) were the most correlated independent variables.

For the panel data model estimation, the dependent variable (Y) was $CO_2$ level, influenced by 6 independent factors: RES ($X_1$), biofuels ($X_2$), bioenergy ($X_3$), population ($X_4$), urbanization ($X_5$), and real GDP per capita ($X_6$).

The panel data model was estimated with the Hausman test. The test quantifies the impact of the independent variables on the dependent variable and the time frame is from 2000 to 2019. The results of the Hausman test could are revealed in Table 5.

**Table 5.** Hausman Test Results.

| Test Summary | | Chi-Square Statistic | Chi-Square D.F. | Probability |
|---|---|---|---|---|
| Random cross-section | | 11.3245 | 8 | 0.1103 |
| Endogenous variable | Exogenous variable | Coefficient | Probability | R_squared |
| CO$_2$ | Renewable energy ($X_1$) | −0.218 | 0.039 | |
| | Biofuels ($X_2$) | −0.182 | 0.041 | |
| | Bioenergy ($X_3$) | −0.125 | 0.037 | 0.5564 |
| | Population ($X_4$) | 0.139 | 0.029 | |
| | Urbanization ($X_5$) | 0.164 | 0.032 | |
| | Real GDP_per_capita ($X_6$) | 0.137 | 0.012 | |

Source: Authors' determined values by using EViews 11.0.

From the Hausman test results, revealed in Table 5, we could observe that the p_value (probability = 0.1103) exceeds the cut-off value of 0.05. That takes us to the conclusion that our model has random effects. Moreover, the panel data quantitative model is well defined and all independent factors in the econometric model are significant. We could also conclude that 55.64% of the variability of the $CO_2$ emission levels in EU countries are described by the factors of the panel data regression model.

Since the parameters of the exogenous variables in the model are statistically significant (Significance less than 0.05) (see Table 5), we could state that all six exogenous variables in the econometric model are statistically significant. In addition, the negative values of $\beta_1$, $\beta_2$, and $\beta_3$ coefficients lead to the conclusion that renewable energy, biofuels, and bioenergy have a positive impact on $CO_2$ rates, which validates all hypotheses from Table 1. Moreover, the positive values of $\beta_4$, $\beta_5$, and $\beta_6$ also validate our assumptions that population, urbanization, and economic growth have negative and significant impact on the increasing rates of $CO_2$ emissions.

The $CO_2$ emission levels in EU countries were analyzed from 2000 to 2019 with a panel data econometric model by independent factors and the model estimation (see Table 5) is: $Y = -0.218X_1 - 0.182X_2 - 0.125X_3 + 0.139X_4 + 0.164X_5 + 0.137X_6$. Thus, it could be concluded that all exogenous variables are strongly correlated with carbon dioxide emission levels. Moreover, while RES, bioenergy, and productivity of energy have positive impacts on lowering the $CO_2$ emissions, any increase of urbanization rate, population, and real GDP per capita would lead to increasing levels of $CO_2$ emissions.

Since the value of the coefficient of determination is 0.5564, we conclude that the model explains 55.64% of the variation of $CO_2$ emission levels in EU member states.

The econometric model is well defined and significant. Moreover, the residuals are not autocorrelated and there are no collinearity problems among the independent variables. The conclusions are in line with other recent papers [37–41] on the nexus between RES and $CO_2$ levels.

Our results validate all six hypotheses and we conclude that bioenergy, RES, energy productivity, urbanization rate, population, and real GDP per capita have a significant impact on $CO_2$ emission levels in EU countries. These confirm other recent studies [42–45], which argue that urbanization rates and population are significant factors for $CO_2$ emission levels.

## 5. Conclusions

In this article, the authors demonstrated that one of the most important factors for $CO_2$ emissions are RES, biofuels, bioenergy, level of urbanization, and population. While renewable energy, biofuels, and bioenergy factors are negatively correlated with $CO_2$ emissions, the other two exogenous factors in the model, population and urbanization, have a strong impact and are correlated with increasing levels of $CO_2$ emission in the EU. Given that the independent factors of the econometric model explain an important share of the $CO_2$ emissions, there are still other variables correlated with carbon dioxide emission levels in EU countries. The results confirm other recent studies on carbon dioxide emissions and renewable energy nexus at the EU level [46–51].

Additionally, both the lack of collinearity and autocorrelation between the exogenous factors demonstrate that the linearity assumption of econometric model is valid.

One limitation of this analysis is that the econometric analysis covers only a twenty-year period of time, due to data availability. Thus, further research should be based on a longer time period and may also use other macroeconomic indicators, such as Industrial Production or Car Registrations.

In terms of recommendations, designing supporting schemes or incentives for RES capacities, both for producers as well for consumers, along with media campaigns focused on increasing the awareness level are needed to be addressed by the national public authorities. Additionally, considering the impact of biofuels and bioenergy on the decarbonization goals, more efforts should be oriented on clarifying the existing legislation at a national level and to develop appropriate policies in order to increase their use. The national plan to achieve the targets of $CO_2$ emissions' reduction should take into consideration the influence of urbanization and population by developing specific means, in appropriate geographical and territorial conditions.

The panel data econometric model for estimating the carbon dioxide levels in the EU between 2000 and 2019 was valid and well defined, and the exogenous variables were important factors with significant impacts on $CO_2$ emissions in all EU countries. This conclusion is underlined by the fact that all regression coefficients were significant and more than 55% of the variance of the $CO_2$ emissions was explained by independent factors.

In the past decade, EU paid increasing attention to the governmental and private society implications of developing low-carbon environments. The policy makers manifested the need for individual implications towards climate change, and the last few years has witnessed increasing implications of the communities in lowering $CO_2$ emission levels in their countries. The transition to a low carbon environment could be attained by state aid schemes, governmental programs, and civil society involvement.

**Author Contributions:** Conceptualization, M.B. and A.C.N.; methodology, M.B.; software, M.B.; validation, M.B. and A.C.N.; formal analysis, M.B.; investigation, A.C.N.; resources, A.C.N.; data curation, A.C.N.; writing—original draft preparation, M.B.; writing—review and editing, A.C.N.; visualization, M.B.; supervision, A.C.N.; project administration, M.B.; funding acquisition, A.C.N. All authors have read and agreed to the published version of the manuscript.

**Funding:** This research received no external funding.

**Data Availability Statement:** Data is contained within the article.

**Conflicts of Interest:** The authors declare no conflict of interest.

## Abbreviations

| | |
|---|---|
| EU | European Union |
| RES | Renewable Energy Sources |
| EViews | Econometric Views |
| Eurostat | European Union Statistical Office |

VIF　　Variance Inflection Factor
GDP　　Gross Domestic Product
$CO_2$　　Carbon Dioxide

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
