# Peer review of "Analyzing the Renewable Energy and CO2 Emission Levels Nexus at an EU Level: A Panel Data Regression Approach"

_processes, doi:10.3390/pr9010130_

Round 1

Reviewer 1 Report

Dear authors,

Thank you for submitting your manuscript for review at Processes. Please consider the following comments.

Summary:

The authors analyze the relationship between CO2 emissions and six factors: renewables, biofuels, bioenergy, population, urbanization, and GDP per capita. They use panel data from the EU to estimate a multiple regression model. Results suggest that renewables, biofuels, and bioenergy are associated with reduced CO2 emissions. Conversely, population, urbanization, and GDP per capita are associated with increases in emissions.

Specific comments:

  • Figure 2: no need to highlight Romania.
  • Line 124: there are six hypotheses.
  • Line 127: your model has six exogenous factors.
  • Equation 1: you should also add subscript “t” since this is a panel data model. For example: (x1)it
  • Line 141: X1 to X6.
  • Line 142: beta1 to beta6.
  • Line 143: clarify why “i” goes up to 28. Is that the total number of countries used?
  • Equation 2: make sure the “i” is subscript.
  • Lines 160, 161: Why are both hypotheses the same? Double-check. One of them has to be different.
  • Line 181: six hypotheses.
  • Table 3: Why is N=28? Shouldn’t it be higher? 28 times the number of years available.
  • Line 223: Which years do you use? In line 69 you say 2000-2019, but here you say 2010-2017. Please clarify and fix it everywhere.
  • Line 226: You cannot directly say which variable has a stronger impact because they have different scales. Unless you are transforming the variables somehow, which is not reported. You should probably just stick to interpreting the signs of the coefficients.

Author Response

Dear professor,

Thank you for all your valuable comments. 

Please see attached our responses.

Kind regards.

Reviewer 2 Report

The revised version looks ok. The paper can be accepted.  

Author Response

Dear professor,

Thank you for all your help during the revision process.

Happy New Year!
